# Convergence Analysis of Path Planning of Multi-UAVs Using Max-Min Ant Colony Optimization Approach

**DOI:** 10.3390/s22145395

**Published:** 2022-07-19

**Authors:** Muhammad Shafiq, Zain Anwar Ali, Amber Israr, Eman H. Alkhammash, Myriam Hadjouni, Jari Juhani Jussila

**Affiliations:** 1Electronic Engineering Department, Sir Syed University of Engineering & Technology, Karachi 75300, Pakistan; muhshafiq@ssuet.edu.pk (M.S.); aisrar@ssuet.edu.pk (A.I.); 2Department of Computer Science, College of Computers and Information Technology, Taif University, P.O. Box 11099, Taif 21944, Saudi Arabia; eman.kms@tu.edu.sa; 3Department of Computer Sciences, College of Computer and Information Science, Princess Nourah Bint Abdulrahman University, P.O. Box 84428, Riyadh 11671, Saudi Arabia; mfhaojouni@pnu.edu.sa; 4HAMK Design Factory, Häme University of Applied Sciences, 13100 Hämeenlinna, Finland; jari.jussila@gmail.com

**Keywords:** path planning, Max-Min Ant Colony Optimization, differential evolution, Cauchy mutation

## Abstract

Unmanned Aerial Vehicles (UAVs) seem to be the most efficient way of achieving the intended aerial tasks, according to recent improvements. Various researchers from across the world have studied a variety of UAV formations and path planning methodologies. However, when unexpected obstacles arise during a collective flight, path planning might get complicated. The study needs to employ hybrid algorithms of bio-inspired computations to address path planning issues with more stability and speed. In this article, two hybrid models of Ant Colony Optimization were compared with respect to convergence time, i.e., the Max-Min Ant Colony Optimization approach in conjunction with the Differential Evolution and Cauchy mutation operators. Each algorithm was run on a UAV and traveled a predetermined path to evaluate its approach. In terms of the route taken and convergence time, the simulation results suggest that the MMACO-DE technique outperforms the MMACO-CM approach.

## 1. Introduction

**Motivation**: Today, the applications of Unmanned Aerial Vehicles (UAVs) in the field of aeronautics are expanding day-by-day, due to their impact in every field [1]. At once, a single UAV was able to do a small task with a high operational cost. With the development of research and technology, multiple UAVs are used for complex tasks e.g., military, construction, surveying, and pattern formations [2,3,4,5]. However, one of the best reasons to use UAVs in a lethal environment is to secure humans from an ambiguous situation [6].

Biological behavior in nature is so inspiring for the real-world problem formulation of aerial robotics [7,8]. When aerial robotics becomes complex in multi-tasking applications, these biological behaviors will help find the optimal solution. In most cases related to the formation and path planning problems of UAVs, various bio-inspired algorithms become feasible [9]. However, one of the oldest optimization techniques widely used for shortest path routing problems is Ant Colony Optimization (ACO) [10,11,12].

**Background and Related Work**: The dynamics of complex aerial systems are difficult to handle especially when they are in clusters and want to achieve the same target smoothly [13]. Therefore, various controlling and optimization techniques are widely used in this area. Some famous intelligent optimizing algorithms, i.e., ACO, PSO, ABC, PIO, etc., are gaining popularity due to their problem-solving ability with the simplest structure [14,15,16,17]. Based on the food searching intelligence of real ants in nature; Ant Colony Optimization and its variants have been used to solve the complex dynamics of a system [18]. ACO initially proposed by Dorigo et al. [19] was applied to Travelling Salesman Problem (TSP). However, the ACO algorithm resolves path planning problems of UAVs to obtain better routes and faster convergence [20].

Rapid progress in this area needs a hybrid approach based on ACO to optimize the previous attainments [21,22,23]. In [24], MAX–MIN Ant System (MMAS), an Ant Colony Optimization technique evolved from Ant System in which search space is bounded for better results. Later on, in [25], the author introduces efficient route planning by achieving the maximum convergence of the target. Similarly, in [26], an improved ACO is used to solve the trajectory planning of multiple UAVs. In [27], Duan et al. combined the ACO algorithm with Differential Evolution (DE) for 3D path planning of uninhabited combat air vehicles. Another author contributed in [28], for feature selection based on the combination of ACO and DE.

**Contributions:** This article compares the convergence rate of two state-of-the-art hybrid algorithms based on Max-Min Ant Colony Optimization techniques for the path planning of multiple UAVs. In [29], the author proposed a hybrid algorithm of Max-Min Ant Colony Optimization combined with Differential Evolution (MMACO-DE) on behalf of the path planning multiple UAVs. The author also added the dynamic environment in his research and proved with the simulation that the target achieved by his proposed algorithm has better results with a successful collision avoidance approach. The main feature of this article is that it selects only the finest ant in each cluster among all for the creation of the required path. In addition, the multiple colonies concept in the research saves the duration of target detection with fewer computations. In [30], the author used the Cauchy Mutant operator along with Max-Min Ant Colony Optimization (MMACO-CM) to improve his previous results. In this article, the same issue regarding path planning of Multiple UAVs in a dynamic environment is resolved with the new hybrid technique of MMACO-CM. This article has two important features, which include increasing the convergence speed for the avoidance falling into local optimum and achieving the shortest path for the target.

**Organization:** The following section is prearranged as follows. Section 2 elaborates the problem statement associated with the path planning of UAVs. Section 3 presents the preliminaries of Unmanned Aerial Vehicles. Section 4 deals with hybrid algorithm along with their mathematical modeling. Section 5 discusses the results obtained by comparison of the algorithm while Section 5 concludes the article.

## 2. Problem Statement

Path planning of aerial vehicles needs precise optimization techniques to obtain optimal routes despite manmade and natural threats. To achieve the target in the shortest possible time along with the shortest distance taken by each UAV, the best hybrid algorithm needs to have minimal complexions. However, the study uses the artificial obstacle theme containing mountains with different peaks and tornados in simulations. Unlike in urban environments, the mountainous area has uneven peaks and hence poses a greater challenge. Moreover, each hybrid algorithm requires a UAV to follow the same path simultaneously. Both UAVs move from the same starting point to the desired location to analyze the performance of the individual algorithm. Each UAV will travel along the specified route by implementing the hybrid algorithm and providing optimal route as well as convergence speed.

## 3. Preliminaries of Unmanned Aerial vehicles

### 3.1. Path Planning

The term “Path Planning” determines the route planned for an object for any specified mission, which includes various obstacles along the path [31]. The planned path restricts the UAV from possible crashes from obstacles or neighboring UAVs. Path planning is more feasible when operating with large quantities of robots. Path planning of aerial systems is categorized into motion-based and tracking-based approaches [32].

*Motion planning:* The term motion planning refers to the movement of aerial or ground robots for the specified or desired task [33]. The basics of motion planning include the shortest possible path, along with a precise turning angle. Motion planning consists of two basic configurations i.e., start and goal configurations. It uses two-dimensional (2D) or three-dimensional (3D) space configurations to show their path [34].

Table 1 shows the difficulty level of robotic motion planning in terms of information provided to the robot. The robot may calculate the size, nature, and distance of obstacles at every instant to avoid a collision [35]. There are four possible scenarios shown in Table 1. The first scenario is the simplest one with completely known information while in the third scenario, the information is partially known. In the second and fourth scenarios, the information regarding obstacles is completely and partially known for dynamic obstacles; therefore, these scenarios are considered difficult ones for robotic motion planning [36].

*Trajectory planning:* Trajectory planning comes into existence when another motional variable encloses the path planning rather than obstacles. The velocity of robots, time taken by the path, and relative kinematics of the system are equally responsible to achieve the goal [37]. Moving from the initial point to the final point using collision avoidance is what trajectory planning is all about. In trajectory planning, both discrete and continuous methods use parametric calculations required to achieve the target. To follow a specific path with control parameters, e.g., location, rate, and acceleration, there must be the scheduling of time, and applying a control system that can accurately execute the trajectory is required [38].

When aerial vehicles fly simultaneously to achieve the desired task, they must follow the planned path to achieve that target. However, artificial intelligence and state-of-the-art technologies are still not capable of providing real-time solutions to UAVs in operating mode [39]. Therefore, nature-inspired algorithms based on the natural behavior of species are serving to provide solutions for the real time scenarios with fewer computations [40].

### 3.2. Collision Avoidance Protocol

Path planning of UAVs is incomplete without using a collision avoidance protocol. When UAVs form a specific shape during flight then they must follow the air collision rules to maintain a safe distance [41]. The environmental hurdles can be of many types including mountain, air disturbance, harsh weather, or any unwanted disturbance. The distance between the UAVs continues to decrease when moving towards the destination due to path planning constraints. To avoid a collision near the target, a safe distance is required by the following relationship [42].
(1)$$L_{\mathrm{safe}}\left(t\right)=\{\begin{array}{c} L,\;t\leq T_{d,n}\\ l,\;t>T_{d,n} \end{array}$$
where L and l are the distance between UAVs and it varies according to the path planning. As a result, when planning or controlling multi-UAV formations, make sure there is no overlap between UAVs as much as possible to build air space cooperation.

Furthermore, changing the height of a UAV regularly puts the flight’s safety at risk. As a result, it is best to avoid changing the altitude regularly. The fluctuating height $C_{h}$ of the UAV states
(2)$$C_{h}=\sqrt{\frac{1}{m_{k}}\overset{m_{k}}{\underset{k=0}{\sum }}{\left(\left(h_{k}-\left(\frac{1}{m_{k}+1}\right)\sum_{l=0}^{m_{k}}h_{k}\right)\right)}^{2}}$$
where $h_{k}$ is the height of UAV, kth represents path leg and leg number is denoted by $m_{k}$. Because all UAVs are flying close together, the turning angle is a significant component of information control and path planning issues. The connection between nodes, edges, and legs is shown in Figure 1. The UAV travels between nodes, which serve as waypoints. It is limited to moving solely between nodes. When the algorithm begins, nodes are created. While the surroundings are three-dimensional, we separate it into the x, y, and z planes to make calculations easier. The nodes are therefore in 2D space. The dynamic topology of these nodes changes depending on the situational context.

Each UAV has various limits in cooperatively altering its attitude due to constraints in the maneuverability of its maximum angle, which may result in an impact between UAVs. The projection of kth and (k+1)th leg on the parallel plan of the present location is presented by $p_{k}$ and $p_{k+1}.$ The calculated spinning angle along with its limited maneuverability is given by ψ.
(3)$$Cos\psi_{max}\leq \left(p_{k}^{T}p_{k+1}/\left|p_{k}\right|\left|p_{k+1}\right|\right),\;k=1,\;2,\;\ldots .,\;n-1$$

### 3.3. Environmental Threats

Air space contains various threats and hurdles for flying vehicles, which can produce uncertainties and delays in-flight operation [43]. In natural threats, air dynamics, weather, humidity, and temperature can disturb UAV flight operations. Similarly, artificial threats including enemies, hurdles, and obstacles can also disturb its operation. To cope with these types of threats, a 3D complex environment needs to be created in simulation [44]. This environment tests the performance of the proposed algorithm by adding uncertainties with respect to the system. In this study, there are two types of environmental threats, which contain multiple mountains with altered peaks and air disturbance in between mountains to create cyclones.

## 4. Hybrid Algorithm

To optimize the system more rapidly and accurately, a hybrid algorithm plays a vital role by combining two or more algorithms. It selects a suitable collection of optimization algorithms to reduce the errors in the system.

### 4.1. Ant Colony Optimization

In the field of optimization algorithms, the Ant Colony Optimization algorithm is extensively used in UAVs. It provides the optimal solution for a complex problem associated with air vehicles. When compared with other algorithms and techniques, ACO has an edge in its distributed computation approach and pre-mature convergence avoidance. In the ant system, real ants create the food-searching algorithm with the help of multiple tours. The most followed route is the optimal route for the rest of the ants. The pheromone trail is the basic tool in the path for indication [45].

Searching for food in insects is common in nature, where each species has their own natural procedure to follow. The most famous species in search of food are ants, due to their unmatched behavior [46,47,48]. The Ant Colony Optimization (ACO) algorithm gives a comprehensive food searching performance of ants [49]. In this algorithm, the social behavior of ants presents mathematically to find the optimal solution for the target. Initially, the ants distribute in different clusters and follow all possible paths to reach the food. In this journey, all ants leave pheromone (a chemical substance) along the route to help other following ants. After the first round, all the ants will try to follow the shortest route in consecutive rounds [50]. Thus, after specific rounds, a single shortest route allowed the ants to follow from the initial point to the target as shown in Figure 2.

In the ACO algorithm, the pheromone value replaces with an updated one after every iteration. This process is autonomous, and all ants will follow the same instructions based on pheromone value, which can reflect the food quality and quantity on the desired path. This process continues until one of the possible paths can get more pheromone than the rest. The preferred path will now be the only path allowed for all ants to form the shortest path in their food search process [10]. However, the path with the most pheromone has a very high probability to be the best path as shown by
(4)$$P_{i,j}^{x}\left(t\right)=\frac{\rho_{i,j}^{\alpha }\left(t\right){ʋ}_{i,j}^{\beta }\left(t\right)}{\sum_{j\in accept(i)}\rho_{i,j}^{\alpha }\left(t\right){ʋ}_{i,j}^{\beta }\left(t\right)}$$
where $P_{i,j}^{x}\left(t\right)$ the probability of xth ant city between i to j, $\rho_{i,j}^{\alpha }\left(t\right)$ is measured pheromone at the corner of the cities, ${ʋ}_{i,j}^{\beta }\left(t\right)$ is the reciprocal of the length between two cities, α and β are the pheromone weight and distance traveled by the xth ant. Initially, the rate of pheromone is not constant at the corners of the cities but later on, the pheromone weight is greater to form a most visible line across all paths [19]. The following relation determines the rate of the pheromone.

(5)$$\rho_{i,j}^{q}\left(t+1\right)=\left(1-\delta \right)\cdot \rho_{i,j}^{q}\left(t\right)+\Delta \rho_{i,j}^{q}$$
where δ ∈ (0,1), the amount of evaporation concerning time *t* to *t*+1
(6)$$\rho_{i,j}^{q}\left(t\right)=\underset{l=1}{\sum }\Delta \rho_{i,j,y}^{q}$$
where $\Delta \rho_{m,j,y}^{q}$, the present amount of pheromone.

### 4.2. Maximum Minimum Ant Colony Optimization

Maximum Minimum Ant Colony Optimization (MMACO) is a technique for improving the system’s search space capability and obtaining the fastest convergence time. The search space is constrained with this strategy by specifying a range, which reduces the search time and allows it to converge quickly. Updated trails in MMACO depend on decent travels of selected ants with the highest fitness value among all ants. In MMACO, consider m ants are assigned for the ith UAV for multiple routes. The mean cost µ of the route gives,
(7)$${µ}_{i,m}\left(t\right)=\frac{1}{m}\overset{m}{\underset{k=1}{\sum }}j_{i,k}\left(t\right)$$

To fulfill the required condition of route cost$\;{µ}_{i,min}\left(t\right)\geq {µ}_{i,k}\left(t\right)$, Hunt stagnation alleviates a distinct solution for the finest iteration global ants for the updated trail pheromone. This form of stagnation avoids the next resolution of the pheromone trails. Although the discrepancy among pheromone trails is prohibited with the minimal effects of pheromone trails.

ACO conducts maximum and minimum pheromone trails. The following equation is responsible to update the pheromone trails in the final iteration of upgrading the trail pheromones. The entire pheromone trail is denoted by $\rho_{max}$ and$\;\rho_{min}$. ACO carries out the maximum and minimum pheromone trails for all of the pheromone trails referred to as $\rho_{max}$ and$\;\rho_{min}$. Now, improving the trail pheromones in the last iteration, the following equation updates the pheromone points.
(8)$$\rho_{i,j}\left(t\right)=\{\begin{array}{c} \;\rho_{max};\;\rho_{i,j}\left(t\right)\geq \rho_{max}\\ \rho_{i,j}\left(t\right);\;\rho_{max}\geq \rho_{i,j}\left(t\right)\geq \\ \rho_{min};\;\rho_{min}\left(t\right)>\rho_{i,j}\left(t\right) \end{array}\rho_{min}$$

#### 4.2.1. Maximum Minimum Ant Colony Optimization with Differential Evolution

Based on the evolutionary process, a meta-heuristic search technique Differential Evolution (DE) solves the optimization issues. A population-based searching procedure improves the system performance for large search spaces. For continuous optimization problems, this adaptable optimization technique will be providing a better solution than others will. Moreover, DE has three core control factors, upon which it is based i.e., operator selection, mutated DE, and crossover DE. In this technique, a random system solution is subsequently magnified using population vectors. It creates the trail alteration first and then connects the trail mutation with the objective mutations to create an updated distinct. It will accept and replace the prior individual with an updated one who has good fitness results.

In the MMACO-DE algorithm, Zain et al. presented a hybrid algorithm that depends on Max-Min ACO and DE algorithms. This algorithm provides improved performance of path planning of multiple UAVs in 3D search space. Moreover, the multi-colonies approach carries out the shortest route to achieve the target with improved convergence speed. There are multiple sub-colonies in the ant colony system in which each sub-colony has its leader whose fitness value is best among all. This ant is responsible for forming and updating the route for other ants to follow. Similarly, the best sub-colony will lead the other colonies to obtain a robust path [24].

To deal with the issues related to path planning, the finest ant from every single colony will represent the colony, and the finest ants of the colony control the pheromone trails for the entire situation. There are three colonies, each with its colony number, which limits the total number of ants, according to the MMACO strategy model. There will be no DE operation if the value of $P_{\mathrm{cr}}$ is zero as shown in Equation (10). Nonetheless, one element of the mutation pheromone mechanism has now been identified which will be delivered to the newly generated pheromone matrix with confidence. However, the DE mutation method and a better trail spreading form a general equation.
(9)$$\rho_{i,j}\left(t\right)=\{\begin{array}{c} \rho_{i,j}^{t+1},\;\mathrm{if}\;t\leq P_{\mathrm{cr}}\\ \rho_{i,j}^{t},\;t>P_{\mathrm{cr}} \end{array}$$
where $\rho_{i,j}^{t+1}$ is denoted as the number of pheromone trails between two repeated nodes i to j of the ant colony and $P_{\mathrm{cr}}$ is crossover probability.

#### 4.2.2. Maximum Minimum Ant Colony Optimization with Cauchy Mutant Operator

To increase the performance of UAVs in difficult situations, one more combinatorial approach for path planning is applied. To improve the system’s performance, Max-Min Ant Colony Optimization pairs with the Cauchy mutation approach. The major perceptive for combining these two algorithms is to speed up convergence and to keep track of any UAV’s future failure. The Cauchy mutation distribution gives the following formula,
(10)$$f\left(a:a_{0},z\right)=1/\pi \left[t/{\left(a-a_{0}\right)}^{2}+t^{2}\right]$$
where Cauchy distribution cost is $a_{0}$, t is the thickness value related to the maximum cost of Cauchy distribution. The incremental function of the distribution function is as follows,
(11)$$F\left(a:0,1\right)=\frac{1}{\pi }\tan^{-1}\left(a\right)+\frac{1}{2}$$

*Compass and map operator:* Compass operators employ conventional ant colony optimization (ACO) to identify the boundaries of the equivalent direction. Furthermore, the map operator utilizes to examine the global best objects to determine the location, velocity, and subsequent transformation of the ACO’s environment to the best individuals. The compass and map operator denotes the coefficient${\;M}_{1}$. It will not help you find the search area, but it might help you reduce the threat level. The Cauchy distribution and its mutation mass coefficient state as follows.
(12)$$\{\begin{array}{c} \mathrm{rand}=\frac{1}{2}+\frac{1}{\pi }\tan^{-1}\left(M_{1}\;\right)\\ M_{1}=\tan\left(\pi \left(\mathrm{rand}-1/2\right)\right) \end{array}$$

The rand function is used in the preceding equation to choose a random value of 0 or 1. The rule develops to update the position of each best ant for each iteration.
(13)$${\hat{X}}_{k}=X_{0k}+M_{1}\left(X_{0k}-X_{\mathrm{gbest}}\right)$$
where ${\hat{X}}_{k},{\;X}_{0k}$ are the current locations of the kth ant with the updated one. Similarly, the global best location is denoted by $X_{\mathrm{gbest}}$. Moreover, the next iteration for the specific location follows,
(14)$$X_{k}=\{\begin{array}{c} f\left({\hat{X}}_{k}\right)<{\hat{X}}_{k},\;f\left(X_{0k}\right)\\ f\left(X_{0k}\right)<X_{0k},\;f\left({\hat{X}}_{k}\right) \end{array}$$

When map and compass operator values of Cauchy mutation become positive, then the position updates the global optimal position. Furthermore, partial distinct entities will fail to discover the optimal and improved location. This happens due to the Cauchy mutation variations. When comparing the unadapted and current position, the superior result is obtained. This approach will not only ensure the optimization method’s supremacy but will also broaden the population’s range [25].

*Landmark operator:* Traditionally, the landmark operator in ACO reduces the population size instantly after every iteration update and a small number of ants will move towards the map’s edge. In an unsuitable manner, this rapid decrease in population will induce an undeveloped convergence of the approach, which results in an unfavorable perception of the optimization at the landmark operator stage. To improve the convergence rate, we must use the Cauchy landmark function instead of the local landmark operator to update the position of each ant’s colony to the best possible place.

The Cauchy landmark function states that.
(15)$$F\left(a:0,1\right)=\frac{2}{\pi }\tan^{-1}\left(a\right)+\frac{1}{2}$$

The landmark operator’s Cauchy mutant weight coefficient is $M_{2}$ and it follows the appropriate distribution is given by;
(16)$$\{\begin{array}{c} \mathrm{rand}=\frac{2}{\pi }\tan^{-1}\left(M_{2}\right)\\ M_{2}=\tan\left(2/\pi *\mathrm{rand}\right) \end{array}$$

The above expression for the updated location is
(17)$$X_{0k}^{N_{c}}=X_{0k}^{N_{c}-1}+M_{2}\left(X_{\mathrm{gbest}}-X_{0k}^{N_{c}-1}\right)$$
where the term $X_{0k}^{N_{c}-1}$ is the distinct kth ant position of $N_{c}-1$ iteration. During the Cauchy mutant landmark operator’s operational stage, all diverse ants will gradually achieve the global ideal result. A suitable Cauchy mutant operator will efficiently move the ant colony at a proper speed and direction, ensuring the algorithm’s stability and speedy convergence.

## 5. Results and Discussion

This section compares the effectiveness of both the algorithms and puts their results side by side to determine which algorithm performs better. The simulations were performed on a computer with an Intel Core i7-1165G7 processor, 16 GB DDR4 Ram, and Windows 11 operating system. The simulation software used was MATLAB 2021a.

The hybrid algorithms of Max-Min ACO apply to two different UAVs to obtain the best result among them in terms of the route followed and convergence rate. In UAV1, MMACO combined with DE to follow the route from the initial point to the target in the presence of a bunch of obstacles. Similarly, UAV2 specifies the effectiveness of the second algorithm i.e., MMACO with CM.

Case 1:

In case 1, the UAVs start from the same initial point and have the same target as well. Wind forces are present in this scenario. Table 2 presents the constraints of the wind force.

Table 3 gives the initial and target points of UAVs along with the total distance traveled by these two for case 1.

To determine the best algorithm for case 1, both UAVs start from the same location simultaneously to the same target point (*) above the ground and are allowed to choose their own shortest path according to their approaches. Figure 3a,b describes the path obtained by UAV1 and UAV2 from the initial point to the target point in a 2-dimensional system. As we can see in Figure 3a, the path constructed by the MMACO-DE is shorter than MMACO-CM and has fewer turns. While Figure 3b, presents the different view of the path followed by the compared algorithms implemented on UAVs.

The 3 dimensional view for case 1 has also been shown in Figure 4 which provides the sense of flying in a dynamic environment including obstacles such as tornados and mountains with different peaks. Again, it is clear from Figure 4 that the MMACO-DE takes fewer turns, hence saving time and minimizing the distance.

The take-off points for both UAVs is (0,2,0) which means that both UAVs will fly simultaneously. Initially, both UAVs will fly on their route by analyzing the obstacles and target point (20,20,1) but after a 2 Km distance, MMACO-CM goes slightly off the route which increases its time of flight whereas MMACO-DE continues to follow the shortest route. At the end of the journey, UAV1 will reach the destination earlier than UAV2, and it verifies that the convergence rate of UAV1 is better than UAV2.

Case 2:

For case 2, the UAVs start from different initial points. Wind forces are present in this scenario as well. The constraints for the wind forces are the same as case 1 and given in Table 2. Table 4 gives the initial and target points of UAVs along with the total distance traveled by these two for case 2.

To determine the best algorithm for case 2, both UAVs start from different locations and are allowed to choose their own shortest path according to their approaches. Figure 5 describes the path obtained by UAV1 and UAV2 from the initial point to the target point in a 2-dimensional system. As we can see in Figure 5a, the path constructed by the MMACO-DE is again shorter than the MMACO-CM and has fewer turns.

The 3-dimensional view for case 2 is shown in Figure 6 which provides the sense of flying in a dynamic environment including obstacles such as wind forces and mountains with different peaks. Again, it is clear from Figure 6 that MMACO-DE takes fewer turns, hence saving time and minimizing the distance.

Analyzing the two case studies, we can clearly see that even if the UAVs start from different locations, MMACO-DE still performs better than MMACO-CM. It picks the shortest distance while avoiding the wind forces and the uneven peaks. It also takes less turns, which in turn ensures that MMACO-DE takes less time to reach the destination than MMACO-CM.

Figure 7 presents the estimation costs of MMACO-DE and MMACO-CM to validate the work.

## 6. Conclusions

Nowadays, bio-inspired algorithms are becoming famous to solve issues related to path planning of unmanned aerial systems with their simplest approach. There are numerous methods in nature offered for this cause i.e., Particle Swarm Optimization, Ant Colony Optimization, Pigeon Inspired Optimization Artificial Bee Colony Optimization, etc., however, one of the most commonly used algorithms for path planning of Unmanned Aerial Vehicles (UAVs) is Ant Colony Optimization (ACO). This article compared two hybrid optimization algorithms for path planning and determines which one is more efficient. Both algorithms use a modified ACO, called Max-Min Ant Colony Optimization algorithm (MMACO), with another algorithm to enhance their effectiveness. The first hybrid algorithm is a combination of the MMACO with the Differential Evolution approach and the second hybrid algorithm is a combination of MMACO with the Cauchy Mutant approach. The MMACO algorithm has tremendous problem-solving skills, especially in complex environments. To reduce noise and disturbance in operation, along with improvement of robustness in the flying, MMACO combines with Differential Evolution and Cauchy Mutant operator. In MMACO–DE, the route followed by UAVs provides the best and shortest route than basic Ant Colony Optimization (ACO); while in MMACO-CM, a flock of UAVs when compared to basic MMACO achieves the optimal route.

Using simulations, this paper concluded that the MMACO-DE algorithm was better than MMACO-CM in achieving a shorter path with less path cost. For future work, we propose to implement the proposed algorithms on hardware and compare the experimental results with the simulations.

## Figures and Tables

**Figure 1 sensors-22-05395-f001:**
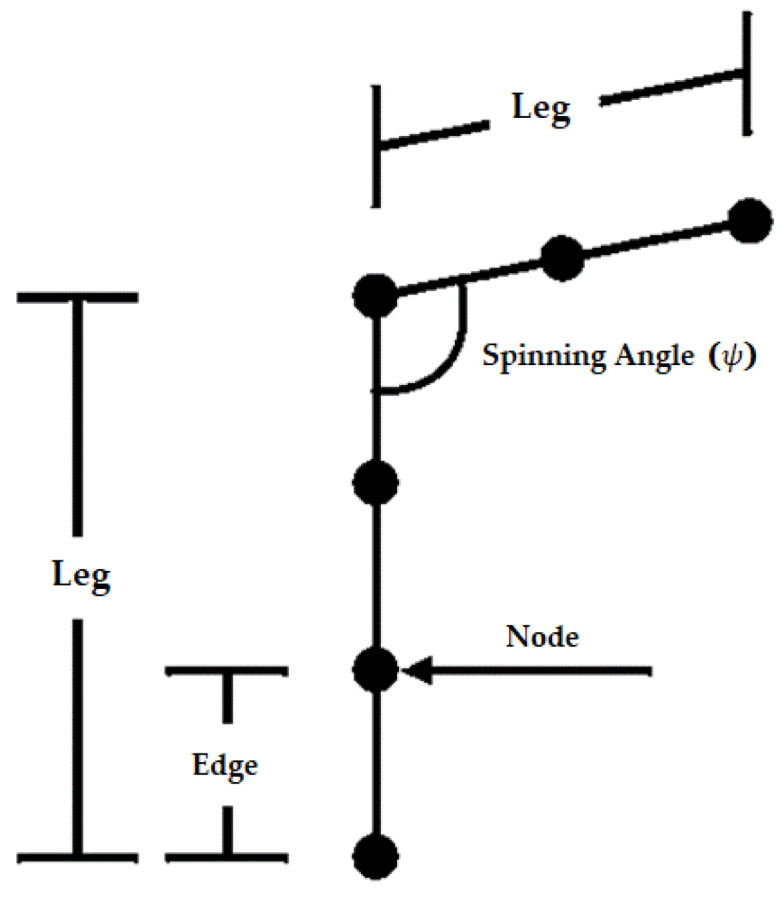
Correlation of Edge, Node, and Legs along with spinning angle ψ.

**Figure 2 sensors-22-05395-f002:**
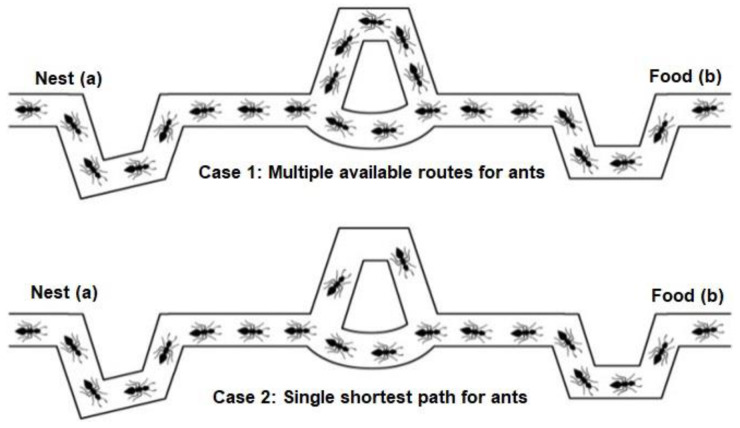
Food hunting procedure in Ant system, case 1 presents the multiple routes while Case 1 shows only single shortest route available for ants.

**Figure 3 sensors-22-05395-f003:**
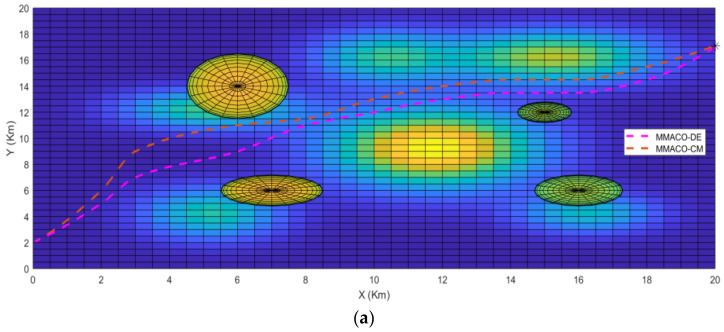
(**a**,**b**) Case1; 2D views of path followed by UAV1(MMACO-DE) and UAV2 (MMACO-CM) to the target (*).

**Figure 4 sensors-22-05395-f004:**
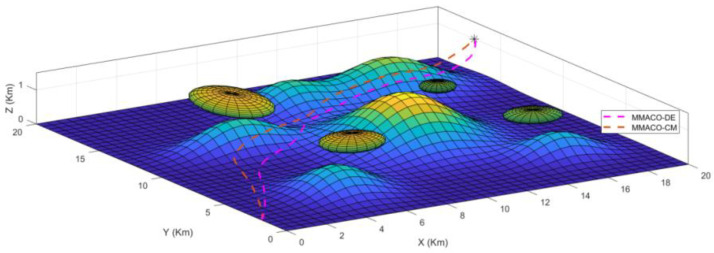
Case1; 3D view of path followed by UAV1 and UAV2 to the target (*).

**Figure 5 sensors-22-05395-f005:**
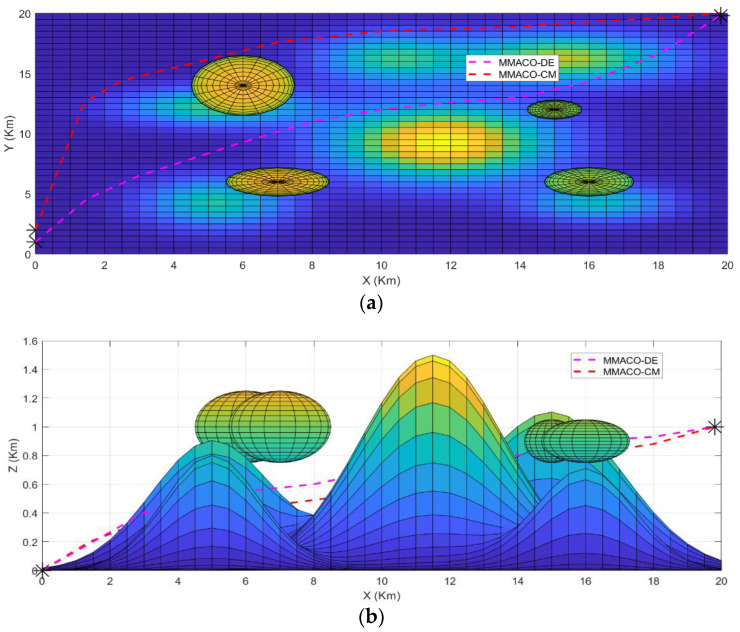
(**a**,**b**) Case 2; 2D path followed by UAV1(MMACO-DE) and UAV2 (MMACO-CM) to the target (*).

**Figure 6 sensors-22-05395-f006:**
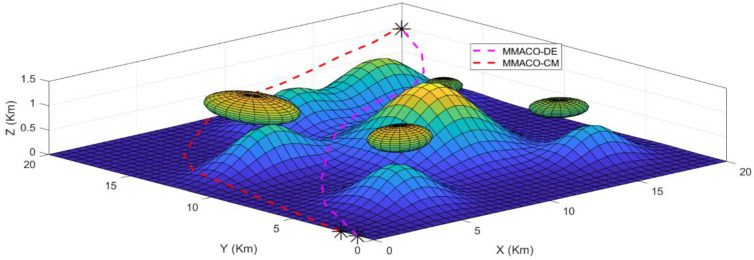
Case 2; 3D view path followed by UAV1 (MMACO-DE) and UAV2 (MMACO-CM) to the target (*).

**Figure 7 sensors-22-05395-f007:**
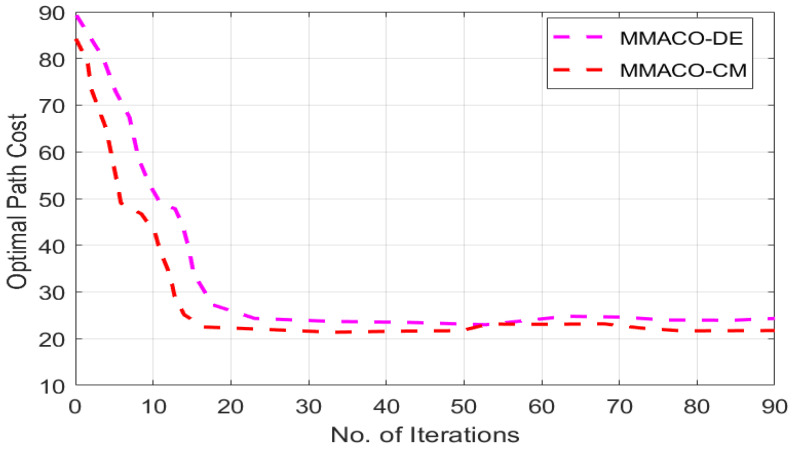
Estimation costs of MMACO-DE and MMACO-CM.

**Table 1 sensors-22-05395-t001:** Possible scenarios between obstacle types and information available.

	Static Obstacle	Dynamic Obstacle
**Complete Information Known**	1st Scenario	2nd Scenario
**Partial Information Known**	3rd Scenario	4th Scenario

**Table 2 sensors-22-05395-t002:** Constraints of wind force.

No.	Constraints	Radius, Center Coordinates	Unit
1	Radius	1.8	km
	Center	(7,6,1)	
2	Radius	2.5	km
	Center	(6,14,1)	
3	Radius	1.3	km
	Center	(16,6,0.9)	
4	Radius	0.8	km
	Center	(15,12,1)	

**Table 3 sensors-22-05395-t003:** Case 1 initial and target points.

UAV	Algorithm Applied	Initial Point (x,y,z)	Target Point (x,y,z)	Distance Travelled (in KM)
UAV1	MMACO-DE	(0,2,0)	(20,20,1)	23.5
UAV2	MMACO-CM	(0,2,0)	(20,20,1)	24.1

MMACO-DE: Max-Min Ant Colony Optimization with Differential Evolution; Max-Min Ant Colony Optimization with Cauchy Mutation.

**Table 4 sensors-22-05395-t004:** Case 2 initial and target points.

UAV	Algorithm Applied	Initial Point (x,y,z)	Target Point (x,y,z)	Distance Travelled (in KM)
UAV1	MMACO-DE	(0,1,0)	(19.8,20,1)	24.221
UAV2	MMACO-CM	(0,2,0)	(19.8,19.8,1)	27.242

MMACO-DE: Max-Min Ant Colony Optimization with Differential Evolution; Max-Min Ant Colony Optimization with Cauchy Mutation.

## Data Availability

All data used to support the findings of this study are included within the article.

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
