# Peer review of "Convergence Analysis of Path Planning of Multi-UAVs Using Max-Min Ant Colony Optimization Approach"

_sensors, 2022, doi:10.3390/s22145395_

Round 1
Reviewer 1 Report
In the paper, the two hybrid optimization algorithms compares for the path planning. The first hybrid algorithm is a combination of the Max-Min Ant Colony Optimization algorithm (MMACO) with the Differential Evolution approach. The second hybrid algorithm is a combination of MMACO with the Cauchy Mutant approach. To reduce noise and disturbance in operation along with improvement of robustness in the flying, MMACO combines with Differential Evolution and Cauchy Mutant operator. Each algorithm was run on a UAV and traveled a predetermined path to evaluate their approaches.
1. The summary and conclusion of the paper need to be further modified and refined to summarize the main innovations and contributions of the paper.
2. In the experiment part, in addition to simulation, it is suggested to add actual experiments. Based on the relevant experimental data, further analysis and summary.
3. It is suggested to add the comparative analysis and verification between the proposed algorithm and other algorithms.
Author Response
Dear Reviewer the compliance reply is attached for your kind consideration.

Reviewer 2 Report
1. Please check the typo error in the manuscript.
2. The language has to be thoroughly revised and rewritten.
3. What is the name of the simulator, and what if the drones fly in urban environments?
4. Why did you consider the obstacle as mountains?
5. Explain the scenarios in clear?
6. Please check figures and equations, because figure-3 is missing and equation numbers are repeated.
7. In the case of the ant algorithm searching for food, the other ants can follow the previous ants, but in the real how the drone can follow the other drones.
Author Response

(The authors gave the same response as above.)

Round 2
Reviewer 2 Report
Equation numbers are the same on both the fourth and sixth pages please change it. Expect this question everything is clear.